# Newborn Screening for Congenital Heart Disease: A Five-Year Study in Shanghai

**DOI:** 10.3390/ijns11020038

**Published:** 2025-05-17

**Authors:** Youping Tian, Qing Gu, Xiaojing Hu, Xiaoling Ge, Xiaojing Ma, Miao Yang, Pin Jia, Jing Zhang, Lulu Yang, Quming Zhao, Fang Liu, Ming Ye, Yulin Yang, Guoying Huang

**Affiliations:** 1National Management Office of Neonatal Screening Project for Congenital Heart Disease, Children’s Hospital of Fudan University, National Children’s Medical Center, Shanghai 201102, China; 2Pediatric Heart Center, Children’s Hospital of Fudan University, National Children’s Medical Center, Shanghai 201102, China; 3Research Unit of Early Intervention of Genetically Related Childhood Cardiovascular Diseases (2018RU002), Chinese Academy of Medical Sciences, Shanghai 201102, China; 4Shanghai Key Laboratory of Birth Defects, Shanghai 201102, China; 5Shanghai Center for Women and Children Health, Shanghai 200062, China

**Keywords:** congenital heart disease, newborn screening, dual-index method, Shanghai

## Abstract

This study aimed to report the progress and results of the newborn screening program for congenital heart disease (CHD) in south Shanghai between 2019 and 2023, and to evaluate the accuracy of the dual-index method (pulse oximetry (POX) plus cardiac murmur auscultation) in clinical practice. Between 2019 and 2023, a total of 198,606 (99.89%) newborns were screened for CHD, of whom 3299 (1.66%) tested positive, 3043 (92.24%) underwent echocardiography for CHD diagnosis and 1109 were diagnosed with CHD in a timely manner. Among 195,307 infants with negative screening results using the dual-index method, 139 (0.07%) were later diagnosed with CHD, and none of these infants died. More than half of these false-negative infants (59.39%) were identified due to the detection of a heart murmur during routine physical examinations within six months after birth. Compared to POX testing alone, the dual-index method significantly improved the sensitivity of screening for CHD, and kept high specificity in clinical practice. This study demonstrated that newborn screening for CHD has been well conducted in Shanghai, and the dual-index method had high accuracy and reliability for neonatal CHD screening in clinical practice.

## 1. Introduction

Congenital heart disease (CHD) can affect approximately 150,000 births per year in China [1], and is a leading cause of infant death [2]. Approximately one-third of these patients have life-threatening major CHD, which requires surgery or invasive intervention within the first year of life to prevent severe complications and death [3]. Almost all life-threatening major CHD cases can be corrected if diagnosed and intervened in a timely manner, while delayed diagnosis is associated with severe complications, including acute cardiovascular collapse, heart failure, hypoxemia-induced acidosis, hypoxic–ischemic brain damage, recurrent pneumonia and infant death [4,5]. Therefore, the early detection of life-threatening major CHD is critical during the neonatal period.

Pulse oximetry (POX) is an easily accessible noninvasive test [6], and has been demonstrated to improve early detection of critical CHD (CCHD), thereby improving outcomes for affected newborns [7]. Our previous studies also demonstrated that POX is an effective screening method for CCHD with moderate sensitivity and high specificity in China, and additionally found that POX plus cardiac murmur auscultation (namely dual-index method) can significantly improve the detection rate of both CCHD and life-threatening major CHD during the early neonatal stage [8,9]. To reduce infant mortality and under-five mortality, the Shanghai Municipal Health Commission implemented newborn screening for CHD using the dual-index method in all birthing facilities, and established a systematic network to cover the entire city with partition management for screening, diagnosis and treatment in 2016 [10].

The Children’s Hospital of Fudan University, a large Grade A tertiary hospital (the highest level in China’s hospital classification system) located in the Minhang District of Shanghai, pioneered the dual-index method and demonstrated its accuracy and feasibility during the birth hospitalization. As one of the four diagnosis and treatment centers for CHD in Shanghai, the Children’s Hospital of Fudan University is responsible for the management, training and quality control of newborn screening program for CHD in the south Shanghai including five administrative districts (Xuhui, Minhang, Songjiang, Jinshan and Qingpu). The primary objective of this study was to update the progress and results of the newborn screening program for CHD in these five districts between 2019 and 2023. And the secondary aim was to evaluate the accuracy of the dual-index method and its continued impact on CHD detection in clinical practice, including an assessment of postnatally diagnosed CHD cases undetected by the dual-index method, over a five-year period in all delivery hospitals in the south Shanghai.

## 2. Materials and Methods

### 2.1. Study Design and Participants

This was a five-year retrospective cohort study involving all live births delivered at hospitals in the five districts of Shanghai from 1 January 2019 to 31 December 2023. The study included all consecutive newborns who participated in CHD screening using the dual-index method, regardless of gestational age or neonatal intensive care unit (NICU) admission status. Newborns who did not receive routine screening (e.g., parent refusal) were excluded from the present study (N = 219).

Before and after delivery, trained medical staff provided education for all participants’ parents and families regarding the risks of CHD, benefits and limitations of newborn screening for CHD, as well as short- and long-term follow-up recommendations. All participants’ parents gave written informed consent for their children before participating in newborn screening for CHD. This study was approved by the ethics committee of the Children’s Hospital of Fudan University.

### 2.2. Procedures

The screening methodology and procedures have been described in detail elsewhere [9,10]. Briefly, screening for CHD was performed in newborn babies aged 6–72 h. Cardiac murmur auscultation was conducted by a trained pediatrician in a quiet room using a stethoscope appropriate for newborns prior to POX screening to ensure that the auscultation result is not affected by noise or the POX result. Then, a trained nurse or pediatrician conducted POX screening using a pulse oximeter with a multisite sensor to measure oxygen saturations from the babies’ right hand and either foot. Dual-index screening for CHD was defined as positive if (a) cardiac murmur is grade II or above, (b) the pulse oximeter oxygen saturation (SpO_2_) is less than 90% in the right hand or either foot, (c) the SpO_2_ is less than 95% in the right hand or either foot on two measurements separated by 2~4 h or (d) a difference between the two extremities is more than 3% on two measurements, separated by 2~4 h. Standardized training and on-site supervision are conducted at least twice each year for medical staff in birthing facilities to ensure the accuracy and reliability of screening results and data registrations [10].

For newborns with positive screening results, clinicians immediately informed their parents of the necessity for echocardiography to rule out CHD. In accordance with the requirements of the Shanghai Newborn Congenital Heart Disease Screening Program [10], newborns who screened positive using the dual-index method were referred to the Children’s Hospital of Fudan University or other diagnosis and treatment centers for echocardiography within one week, and those diagnosed with CHD were further evaluated by pediatric cardiac experts. For infants with negative screening results, clinical follow-up and physical examinations were conducted at local community hospitals at 6 weeks, 3 months and 6 months of age, in combination with feedbacks from parents about cardiac symptoms such as cyanosis, tachypnea and feeding difficulty or medical chart review. Newborns diagnosed without CHD will stay in the track of Child Health Management, a basic public health service program for children under seven years of age in China [10].

During the COVID-19 pandemic, Shanghai implemented a range of effective policies (e.g., nucleic acid testing, vaccination availability, standard infection control measures) to ensure the safety of pregnant and postpartum women during childbirth, balancing infection control with the continuity of essential maternity care [11,12].

### 2.3. Data Source

The Information Management System of Newborn Screening for CHD (www.nchd.org.cn accessed on 16 May 2025) was used daily to register and track the results of CHD screening, diagnosis, and treatment. All delivery hospitals registered information on maternal and neonatal demographic characteristics (e.g., maternal name, maternal resident identity card, infant name, sex and birth date) and screening results (including the screening date, SpO_2_ values and grade of cardiac murmur). The Children’s Hospital of Fudan University uploaded echocardiography, and registered the results of diagnosis (including the diagnosis date, and CHD classification) and treatment (including the date, method and outcome of surgery or interventional catheterization) in this electronic registration system.

### 2.4. Defining Outcomes

As suggested by previous studies [8,13], CHD were classified as critical (requiring intervention or causing death within 28 days of age), serious (requiring intervention before 1 year of age, but not classified as critical), significant (defects persist longer than 6 months of age, but not classified as critical or serious), and non-significant (defects are not physically appreciable and do not persist after 6 months of age). For the purpose of the study analyses, we defined critical and serious CHD as life-threatening major CHD, while the remaining categories were classified as minor CHD.

In this study, the following conditions were excluded from the definition of CHD: (1) patent ductus arteriosus that closed spontaneously within 3 months, (2) atrial septal defect (ASD)  < 5 mm in diameter at 3 months of age, (3) physiological pulmonary branch stenosis that resolved during follow-up, (4) pulmonary stenosis or aortic stenosis with a pressure gradient of <20 mmHg without further deterioration during follow-up and (5) simple patent foramen ovale [3].

### 2.5. Statistical Analyses

The results of newborn screening for CHD in each district per year are presented using the following key indicators: (1) screening rate = number of newborns screened/number of live births × 100%; (2) screened positive rate = number of newborns screened positive/number of newborns screened × 100%; (3) echocardiography examination rate = number of echocardiography in newborns screened positive/number of newborns screened positive × 100%; (4) diagnostic rate of CHD in newborns screened positive = number of newborns diagnosed with CHD in those screened positive/number of newborns screened positive × 100%; (5) CHD prevalence = number of CHD cases/number of live births × 100%.

We calculated the sensitivity, specificity, false-positive rate, positive and negative predictive value, consistency rate, Youden’s index and positive and negative likelihood ratios for both the dual-index method and POX alone in newborn screening for CHD. The Wilson method was used to calculate the 95% confidence interval (95% CI) of sensitivity and specificity. Additionally, we described the false-negative results of the dual-index method. All statistical analyses were performed using SAS 9.4 (SAS Institute Inc., Cary, NC, USA).

## 3. Results

### 3.1. Results of Newborn Screening for CHD in South Shanghai

From 1 January 2019 to 31 December 2023, a total of 198,606 (99.89%) newborns were screened for CHD using the dual-index method across the five districts of south Shanghai (Table 1 and Figure 1). The annual CHD screening rate has exceeded 99% in all districts (Figure 2). Among the 198,606 newborns screened, 3299 (1.66%) tested positive, of whom 3043 (92.24%) underwent echocardiography for CHD diagnosis (Table 1). The echocardiography examination rate increased from 85.63% in 2019 to 97.51% in 2023 (Table 1), demonstrating an upward trend in nearly all districts (Figure 2).

A total of 1109 (33.62%) screen-positive newborns were diagnosed with CHD, including 73 CCHD, 133 serious CHD and 903 minor CHD (Figure 1). As presented in Table 2, CCHD cases included pulmonary atresia (n = 20), total anomalous pulmonary venous drainage (n = 16), coarctation of the aorta (n = 11), transposition of the great arteries (n = 10), critical pulmonary stenosis (n = 7), interrupted aortic arch (n = 3), double outlet right ventricle (n = 2), hypoplastic left heart syndrome (n = 2) and single ventricle (n = 2). Serious CHD cases included ventricular septal defect (n = 71), Tetralogy of Fallot (n = 20), pulmonary stenosis (n = 18), patent ductus arteriosus (n = 15), atrioventricular septal defect (n = 6), coarctation of the aorta (n = 2) and partial anomalous pulmonary venous connection (n = 1).

### 3.2. CHD in Infants with Negative Screening Results of the Dual-Index Method

Among 195,307 infants with negative screening results from the dual-index method, 139 (0.07%) were subsequently diagnosed with CHD. More than half of these false-negative infants (59.39%) were identified due to the detection of a heart murmur during routine physical examinations within six months after birth. The remaining cases were diagnosed through echocardiography, which was arranged for conditions such as preterm birth, low birth weight, pneumonia or other medical concerns.

Fortunately, among these 139 infants, none were diagnosed with CCHD, and 33 infants were diagnosed with serious CHD requiring surgery before the age of one year. These cases included patent ductus arteriosus (n = 17), ventricular septal defect (n = 12), atrioventricular septal defect (n = 1), coarctation of the aorta (n = 1), cor triatriatum (n = 1) and partial anomalous pulmonary venous connection (n = 1, Table 2). There were no deaths among infants with negative screening results using the dual-index method.

### 3.3. Accuracy of Screening Methods for CCHD and Life-Threatening Major CHD

The accuracy of POX and the dual-index method for the detection of CCHD, life-threatening major CHD, and all types of CHD is presented in Table 3. In this retrospective cohort, POX alone as a screening method was able to identify 62 of 73 (84.93%) cases of CCHD, but only 91 of 239 (38.08%) cases of major CHD. The specificity of POX was 99.57% and 99.59% for CCHD and life-threatening major CHD, respectively. The dual-index method could detect 73 of 73 (100.00%) cases of CCHD and 206 of 239 (86.19%) cases of life-threatening major CHD. Compared to POX testing alone, the dual-index method significantly improved the sensitivity of screening for CCHD and life-threatening major CHD. Additionally, the dual-index method could achieve similarly high specificities (CCHD: 98.38%, major CHD: 98.44%), and had similarly high consistency rates.

The positive predictive value of POX was 6.79% for CCHD and 9.97% for life-threatening major CHD, while the positive predictive value of the dual-index method was 2.21% for critical CHD, and 6.24% for life-threatening major CHD.

## 4. Discussion

Using real-world data from a five-year clinical practice, this study detailed the progress and results of the newborn screening program for CHD in the five administrative districts of south Shanghai between 2019 and 2023, and evaluated the accuracy and reliability of the dual-index method and its continued impact on CHD detection in clinical practice. Nearly all live births in these five districts were screened for CHD using the dual-index method, and the echocardiography examination rate and diagnostic rate of CHD in newborns screened positive have reached 92.24% and 33.62%, respectively. Furthermore, this study confirmed that the dual-index method had high sensitivities and specificities for detecting both CCHD and life-threatening major CHD in clinical practice.

Between 2019 and 2023, almost all newborns were screened for CHD using the dual-index method in the five administrative districts of south Shanghai, and the annual screening rate for CHD has exceeded 99% in all five districts. The mean screened positive rate (1.66%) was consistent with findings from our previous multicenter screening studies [8,9]. The echocardiography examination rate among newborns who screened positive in the south Shanghai was 92.24%, showing an upward trend from 85.63% in 2019 to 97.51% in 2023, which was higher than the citywide average (89.88%) between 2017 and 2021 [10]. The overall prevalence of CHD in the five districts (6.28‰) was slightly lower than the prevalence (7.93‰) reported in our previous multicenter prospective screening study conducted at 15 birthing facilities in Shanghai, while the prevalence of life-threatening major CHD (1.20‰) in this study was comparable to that of our previous study (1.21‰) [9]. These findings suggest that while some CHD may be missed or underreported in clinical practice using the dual-index method, life-threatening major CHD cases are consistently identified. Overall, the high screening rate, echocardiography examination rate and the fact that 202 children with CHD received timely and effective treatment indicated that the newborn screening program for CHD has been well implemented in the five districts of south Shanghai and a sound integrated network for neonatal CHD screening, diagnosis and treatment has been established. These findings also underscored that, despite the challenges to the Shanghai healthcare system posed by COVID-19 [14], the newborn screening program for CHD effectively adapted to maintain high standards of care, ensuring the early detection and timely treatment of CHD in newborns.

In our study, the post-discharge diagnosis rate of CCHD was 4/100,000, which aligns with the results from our previous screening study in Shanghai [9] and a UK regional study [15], but lower than the rate in the rest of China [3,8]. The observed differences in the prevalence of CCHD may be attributed to a combination of advanced prenatal diagnosis and regional healthcare disparities. In Shanghai, prenatal diagnosis for CHD is a routine part of pregnancy management. A study by Zhang et al. (2020) found that regions like Zhejiang, which share a similar healthcare infrastructure to Shanghai, reported 90% prenatal detection rates for CCHD [16]. Additionally, genetic studies in Shanghai have shown a high incidence of chromosomal abnormalities in fetuses with CCHD [17], which may influence parental decisions regarding pregnancy termination [18]. Higher maternal education and access to advanced healthcare in Shanghai further contribute to the lower CCHD prevalence. [19].

POX screening for CCHD has been widely recommended and adopted in many countries [20]. In the present study, we found that POX had moderate sensitivity (84.93%) and high specificity (99.57%) for CCHD, consistent with our previous multicenter screening studies [8,9], and with findings from studies conducted in developed countries [6]. However, POX alone as a screening method was only able to identify only 38.08% of cases of major CHD. The sensitivity of POX screening for major CHD in this study was lower than that in our previous studies conducted at 15 birthing facilities in Shanghai (44.3%) [9], and 18 hospitals in 10 provinces of China (58.7%) [3], but higher than a 5-year observational study from the UK (33%) [15]. This study, along with previous research, suggested that POX alone as a screening method may lead to missed diagnoses of many major CHD cases, such as large ventricular septal defects and tetralogy of Fallot [21,22]. Therefore, all major CHD should be regarded as targets for newborn screening to prevent heart failure or irreversible pulmonary vascular disease.

In the present study, the sensitivity and specificity of the dual-index method was 100.00% and 98.38% for CCHD, which were consistent with the findings of our previous multicenter screening studies [8,9]. Although the sensitivity of the dual-index method for major CHD (86.19%) in this study was slightly lower than that in our previous studies [8,9], our findings demonstrated that the dual-index method could significantly improve the sensitivities for CCHD and major CHD compared to using POX testing alone. Additionally, the false-positive rates of the dual-index method for the detection of CCHD and major CHD remained at a low level. These findings provide valuable insights that address concerns in countries like the UK about the missed diagnosis of major CHD using only POX testing and the high false-positive rate. It is also noteworthy that the positive predictive value of the dual-index method for all types of CHD exceeded one-third, indicating that over 30 out of 100 infants with a positive screening result using the dual-index method would be diagnosed and treated in a timely manner. Infants with early detection of CHD may benefit from subsequent timely intervention and intensive surveillance [10]. Importantly, only 0.08% (139 of 195,307) of infants with negative screening results using the dual-index method were later diagnosed with CHD, of whom 33 were identified with major CHD, and there were no deaths among these infants. Over half of these false-negative infants (59.39%) were found to have a heart murmur during routine physical examination within six months after birth, suggesting that infants with negative screening results using the dual-index method should be aware of follow-up physical examination.

Our study has several strengths. First, the present study utilized real-world data from a five-year clinical practice across all delivery hospitals in the five administrative districts of Shanghai, which enhances the applicability of the results to everyday clinical practice. Second, our study comprehensively described the postnatal diagnoses of all types of CHD in infants with both positive and negative results using the dual-index method over a five-year period, and described what cardiac conditions lead to false-negative results using the dual-index method. Third, this study benefited from a relatively large sample size, enabling us to assess whether the accuracy of the dual-index method in clinical practice aligns with findings from our previous large-scale, prospective, multicenter screening studies. Inevitably, the present study also has several limitations. First, as a retrospective database study based on real-world clinical practice, it may be constrained by the lack of direct control and the inherent heterogeneity of the population. Second, 139 infants with negative screening results using the dual-index method were later diagnosed with CHD through clinical follow-up and medical chart review, which indicated that there was a possibility of false-negative results. Third, the low prevalence of CCHD in the current study would be expected to impact the positive predictive value of screening. Future studies should further validate the accuracy of the dual-index method in screening for CCHD in clinical practice.

## 5. Conclusions

In the present study, we detailed the progress and results of the newborn screening program for CHD in the five administrative districts of south Shanghai between 2019 and 2023, and demonstrated that newborn screening program for CHD has been effectively implemented. Additionally, this retrospective cohort study based on clinical practice further confirmed that the dual-index method had high sensitivity and specificity, and a low false-positive rate. Our study suggests that the dual-index method could be applied in common hospital settings in other regions, and that infants with negative screening results should be aware of follow-up physical examination.

## Figures and Tables

**Figure 1 IJNS-11-00038-f001:**
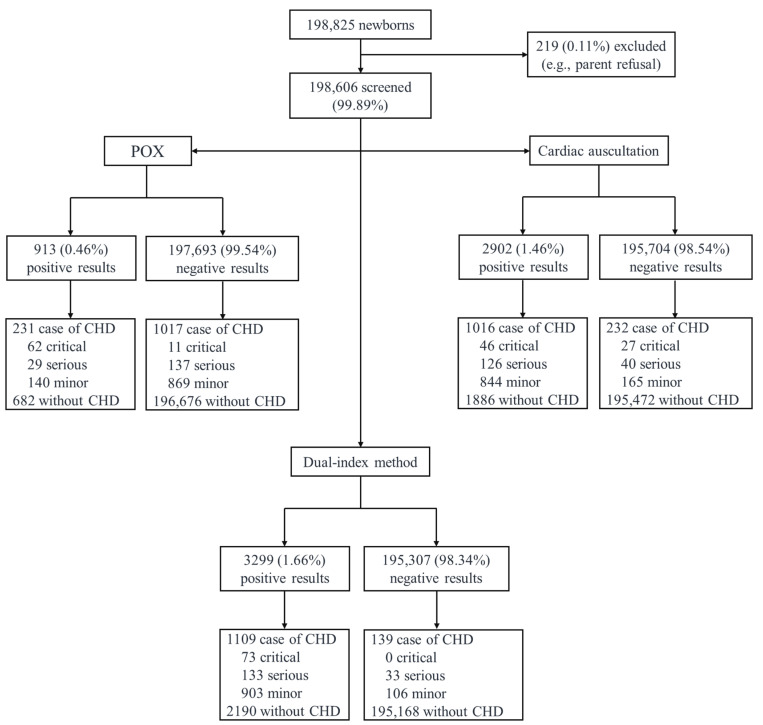
Profile of newborn screening for congenital heart disease in the five districts of Xuhui, Minhang, Songjiang, Jinshan and Qingpu in Shanghai. Note: CHD, congenital heart disease; POX, pulse oximetry.

**Figure 2 IJNS-11-00038-f002:**
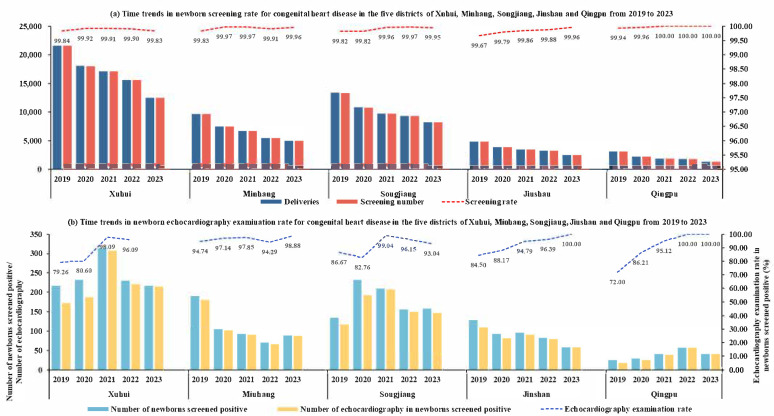
Time trends in newborn screening rate and echocardiography examination rate for congenital heart disease in the five districts of Xuhui, Minhang, Songjiang, Jinshan and Qingpu from 2019 to 2023.

**Table 1 IJNS-11-00038-t001:** Results of the newborn screening program for CHD in the five districts of Xuhui, Minhang, Songjiang, Jinshan and Qingpu in Shanghai from 2019 to 2023.

District	Number of Newborns	Number of Newborns Screened	Screening Rate (%)	Number of Newborns Screened Positive	Screened Positive Rate (%)	Number of Echocardiography in Newborns Screened Positive	Echocardiography Examination Rate (%)	Number of Diagnosed CHD in Those Screened Positive	Diagnostic Rate of CHD in Newborns Screened Positive (%)	Number of False-Negative Cases	CHD Cases	Prevalence of CHD (‰)	Number of Treatments
Xuhui	84,884	84,783	99.88	1210	1.43	1103	91.16	463	38.26	26	489	5.76	60
Qingpu	10,304	10,301	99.97	193	1.87	180	93.26	56	29.02	4	60	5.82	15
Jinshan	17,925	17,891	99.81	459	2.57	420	91.50	110	23.97	27	137	7.64	27
Minhang	34,262	34,235	99.92	547	1.60	527	96.34	220	40.22	41	261	7.62	37
Songjiang	51,450	51,396	99.90	890	1.73	813	91.35	260	29.21	41	301	5.85	63
Total	198,825	198,606	99.89	3299	1.66	3043	92.24	1109	33.62	139	1248	6.28	202

Note: (1) screening rate = number of newborns screened/number of newborns × 100%; (2) screened positive rate = number of newborns screened positive/number of newborns screened × 100%; (3) echocardiography examination rate = number of echocardiography in newborns screened positive/number of newborns screened positive × 100%; (4) diagnostic rate of CHD in newborns screened positive = number of newborns diagnosed CHD in those screened positive/number of newborns screened positive × 100%; (5) prevalence of CHD = number of CHD cases/number of live births × 100%. CHD, congenital heart disease.

**Table 2 IJNS-11-00038-t002:** Types of life-threatening major congenital heart disease in infants with positive and negative screening results using the dual-index method.

CHD in Infants with Positive Screening Results		CHD in Infants with Negative Screening Results	
Critical CHD (N = 73)		Serious CHD (N = 33)	
Pulmonary atresia	20	Patent ductus arteriosus	17
Total anomalous pulmonary venous drainage	16	Ventricular septal defect	12
Coarctation of the aorta	11	Atrioventricular septal defect	1
Transposition of the great arteries	10	Coarctation of the aorta	1
Critical pulmonary stenosis	7	Cor triatriatum	1
Interrupted aortic arch	3	Partial anomalous pulmonary venous connection	1
Double outlet right ventricle	2		
Hypoplastic left heart syndrome	2		
Single ventricle	2		
Serious CHD (N = 133)			
Ventricular septal defect	71		
Tetralogy of Fallot	20		
Pulmonary stenosis	18		
Patent ductus arteriosus	15		
Atrioventricular septal defect	6		
Coarctation of the aorta	2		
Partial anomalous pulmonary venous connection	1		

Note: CHD, congenital heart disease.

**Table 3 IJNS-11-00038-t003:** Accuracy of screening methods for detecting critical and major congenital heart disease, and all types of congenital heart disease (N = 198,606).

	CCHD		Major CHD		All Type of CHD	
	POX Alone	Dual-Index Method	POX Alone	Dual-Index Method	POX Alone	Dual-Index Method
True-positives	62	73	91	206	231	1109
False-negatives	11	0	148	33	1017	139
False-positives	851	3226	822	3093	682	2190
True-negatives	197,682	195,307	197,545	195,274	196,676	195,168
False-positive rate (%)	0.43	1.62	0.41	1.56	0.34	1.11
Sensitivity (%)	84.93 (75.00, 91.37)	100.00 (95.00, 100.00) *	38.08 (32.15, 44.37)	86.19 (81.24, 90.00) *	18.51 (16.45, 20.76)	88.86 (87.00, 90.49) *
Specificity (%)	99.57 (99.54, 99.60)	98.38 (98.32, 98.43)	99.59 (99.56, 99.61)	98.44 (98.39, 98.49)	99.65 (99.63, 99.68)	98.89 (98.84, 98.94)
Positive predictive value (%)	6.79 (5.33, 8.61)	2.21 (1.76, 2.77)	9.97 (8.19, 12.08)	6.24 (5.47, 7.12)	25.30 (22.59, 28.22)	33.62 (32.02, 35.25)
Negative predictive value (%)	99.99 (99.99, 100.00)	100.00 (100.00, 100.00)	99.93 (99.91, 99.94)	99.98 (99.98, 99.99)	99.49 (99.45, 99.52)	99.93 (99.92, 99.94)
Consistency rate (%)	99.57 (99.54, 99.59)	98.38 (98.32, 98.43)	99.51 (99.48, 99.54)	98.43 (98.37, 98.48)	99.14 (99.1, 99.18)	98.83 (98.78, 98.87)
Youden’s index	0.85	0.98	0.38	0.85	0.18	0.88
Positive likelihood ratio	197.91	61.54	91.88	55.28	53.56	80.08
Negative likelihood ratio	15.13	0.00	62.18	14.03	81.77	11.26

Note: CHD, congenital heart disease; POX, pulse oximetry; major CHD includes critical CHD and serious CHD. * Statistically significant (*p* < 0.05) using McNemar’s test compare differences in specificity between POX and the dual-index method.

## Data Availability

The datasets used and analyzed in this study are available from the corresponding author on reasonable request.

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
