# Peer review of "Newborn Screening for Congenital Heart Disease: A Five-Year Study in Shanghai"

_2409-515X, 2025, doi:10.3390/ijns11020038_

Round 1

Reviewer 1 Report

Comments and Suggestions for Authors

This article is well written but I am concerned about some of the issues in the larger context.

In addition to this work, the senior author has published extensively on this topic (2014-Lancet, 2017-Pediatrics, 2019- Journal of Pediatrics, 2023-Lancet).   (These are only those I am familiar with.)

The 2023 work includes babies in Shanghai from 2017-2021 and the current study babies in Shanghai from 2019-2023.  The 2023 Lancet paper is not referenced in the manuscript under review and the reasoning for the over-lap between these populations is not explained.   

Additionally, there appears a significant difference in the prevalence of critical congenital heart disease (CCHD) diagnoses between Shanghai and the rest of the country which is not discussed. 

In the 2014 Lancet study including babies from across China, the prevalence of the 12 CCHD diagnoses used by the American Academy of Pediatrics was just under 12 per 10,000 births.   In the 2019 Journal of Pediatrics article the prevalence was closer to 19 per 10,000. 

However, in the 2017 Pediatrics and 2023 Lancet papers, the prevalence was less than 4 per 10,000.  Although in this work the prevalence was just under 7 per 10,000.

The prevalence of single ventricle heart disease is strikingly low in the three papers from Shanghai at 0.42 (2017), 0.04 (2023) and 0.31(2025) per 10,00.

In any screening test, the pretest probability is of crucial importance in understanding the test's performance.   No mention of prenatal diagnoses or termination of pregnancy is made in this manuscript.  The authors should expound on the unique nature of childbirth in Shanghai that leads to such low rates of critical congenital heart disease and specifically single ventricle heart disease.

I was concerned by the methodology where the goal was to have babies who failed to pass their dual-index screening were referred for echocardiography within seven days.   An important number of babies with CCHD will become symptomatic or succumb to their CCHD before seven days of age.

This manuscript has important information for the congenital heart disease community, but in its present state, that information cannot be understood and is not generalizable to other settings.

Author Response

Reviewer #1: This article is well written but I am concerned about some of the issues in the larger context.

  1. In addition to this work, the senior author has published extensively on this topic (2014-Lancet, 2017-Pediatrics, 2019- Journal of Pediatrics, 2023-Lancet). (These are only those I am familiar with.) The 2023 work includes babies in Shanghai from 2017-2021 and the current study babies in Shanghai from 2019-2023. The 2023 Lancet paper is not referenced in the manuscript under review and the reasoning for the over-lap between these populations is not explained.

Response: Thanks for your comments and suggestions. We appreciate the opportunity to clarify the relationship between our study and the 2023 Lancet paper (Impact of Newborn Screening Programme for Congenital Heart Disease in Shanghai: A Five-Year Observational Study in 801,831 Newborns. The Lancet Regional Health - Western Pacific. 2023). The Lancet paper detailed the implementation and overall progress of the Newborn Screening Program for CHD in Shanghai from 2017 to 2021, with a focus on the impact of the program on infant mortality rates post-implementation. While there is partial overlap in the populations due to the same newborn screening program for CHD, the timeframes and study scopes differ, and the objectives of the two studies are distinct.

This study specifically aims to report the progress of the CHD screening program in the south Shanghai (our hospital’s catchment area), updating data from 2019 to 2023 (Page 2, Line 66-68). Additionally, a major focus of our research is evaluating the continued impact of the dual-index method (POX plus cardiac murmur auscultation) on CHD detection in clinical practice, including an assessment of postnatally diagnosed CHD cases undetected by the dual-index method (Page 2, Line 68-71). Importantly, our current analysis provides a more in-depth evaluation of false-negative cases identified through the dual-index method, with an emphasis on which cases were missed and how these cases were detected post-discharge. This research underscores the importance of using the dual-index method rather than relying on POX alone and highlights critical symptoms to be aware of for early detection.

We have now added the 2023 Lancet reference to the manuscript (Page 2, Line 58) and clarified these differences in objectives and population overlap (Page 2, Line 62-68). Thank you again for your valuable feedback.

  1. Additionally, there appears a significant difference in the prevalence of critical congenital heart disease (CCHD) diagnoses between Shanghai and the rest of the country which is not discussed. In the 2014 Lancet study including babies from across China, the prevalence of the 12 CCHD diagnoses used by the American Academy of Pediatrics was just under 12 per 10,000 births. In the 2019 Journal of Pediatrics article the prevalence was closer to 19 per 10,000. However, in the 2017 Pediatrics and 2023 Lancet papers, the prevalence was less than 4 per 10,000. Although in this work the prevalence was just under 7 per 10,000. The prevalence of single ventricle heart disease is strikingly low in the three papers from Shanghai at 0.42 (2017), 0.04 (2023) and 0.31(2025) per 10,00. In any screening test, the pretest probability is of crucial importance in understanding the test's performance. No mention of prenatal diagnoses or termination of pregnancy is made in this manuscript. The authors should expound on the unique nature of childbirth in Shanghai that leads to such low rates of critical congenital heart disease and specifically single ventricle heart disease.

Response: Thank you for your insightful question regarding the observed low prevalence of critical congenital heart disease (CCHD) in Shanghai compared to the rest of China, likely reflecting the unique nature of prenatal care and childbirth patterns in Shanghai. One important contributing factor is the widespread uptake and high quality of prenatal diagnosis. Prenatal diagnosis for CHD is a routine part of pregnancy management in Shanghai. A study by Zhang et al. (2020) highlighted that regions like Zhejiang, which share similar healthcare infrastructure with Shanghai, reported 90% prenatal detection rates for CCHD. Further, a prenatal genetic study conducted in Shanghai on fetuses with single ventricle and single atria phenotypes reported a high incidence of chromosomal abnormalities and pathogenic copy number variants, factors that likely influenced parental decisions regarding termination of pregnancy in cases of CCHD (Tomek et al. JAMA Netw Open. 2023). Additionally, urban centers like Shanghai often have higher maternal educational levels and greater access to advanced healthcare services in China. A systematic review of CHD in China found that urban areas typically have lower prevalence of CHD due to better screening and prenatal care, while rural areas report higher birth prevalence due to less access to these services (Liu et al., International Journal of Epidemiology, 2019). We have revised our manuscript to include a more detailed discussion on the role of prenatal diagnosis and termination practices in Shanghai, and how they contribute to the observed differences in CCHD prevalence (Page 8-9, Line 287-300).

We agree that pretest probability is a key factor in interpreting screening effectiveness. The lower prevalence of CCHD would be expected to impact the positive predictive value of screening and could influence comparisons across regions with different epidemiologic profiles. We have added it as one limitation in our manuscript (Page 10, Line 357-360).

  1. I was concerned by the methodology where the goal was to have babies who failed to pass their dual-index screening were referred for echocardiography within seven days. An important number of babies with CCHD will become symptomatic or succumb to their CCHD before seven days of age.

Response: Thanks for your comments and suggestions. We understand your concern regarding the seven-day timeframe for echocardiography in newborns who tested positive, especially considering that some babies with critical CHD may become symptomatic or even succumb to the condition before seven days of age. The seven-day referral window, as stipulated by the Shanghai Municipal Health Commission for the Newborn Screening Program for Congenital Heart Disease, serves as a ​​public health policy framework​​ to standardize follow-up timelines across all participating institutions (Page 3, Line 103-108). This policy ensures systematic data collection and quality control for citywide screening efforts. In fact, all infants who screened positive were ​​immediately informed​​ by clinicians about the necessity of echocardiography to rule out critical CHD. We have added it in the method section (Page 3, Line 103-104).

  1. This manuscript has important information for the congenital heart disease community, but in its present state, that information cannot be understood and is not generalizable to other settings.

Response: Thank you for your thoughtful suggestions. We appreciate your recognition of the importance of our research for the CHD community. We understand your concern that, in its current form, the information in our manuscript may not be fully understood or generalizable to other settings. In response to your suggestion, we have made significant revisions to our manuscript to provide clearer context, more comprehensive explanations, and to better highlight the broader applicability of our findings.

We hope that these revisions address your concerns and make our research more accessible and relevant to the broader CHD community. Ultimately, we aim to share our findings and promote the use of dual-index method for CHD screening in other countries and regions, to improve early detection and outcomes for newborns with CHD.

Reviewer 2 Report

Comments and Suggestions for Authors

This report describes newborn screening for congenital heart disease (CHD) from 2019-2023 using pulse-oximetry and the physical exam for murmurs.  This is referred to as the “dual-index method.”  Much of this report lumps together all types of CHD.  However, increasing the focus on CCHD would make it line up better with other studies in the field and focus on those who really benefit from early detection. The distinction is addressed later in the report, but this should be woven in throughout.

Historically, pulse-oximetry was added to the physical exam, which had been the standard of care, to identify critical CHD (CCHD).  Using the term “dual-index” seems to complicate matters.  However, this could be justified if there was something special about the exam for murmurs (e.g., a standardized approach with little inter- and intra-observer variability). 

It is unclear what this study adds to what has already been described about newborn screening for CCHD. One missed opportunity is to describe implementation issues related to the COVID-19 pandemic.  Given the restrictions, I am sure that this important work was challenging.  There could be important lessons.  The discussion makes describing the progress; however, the analysis is more about the overall findings. 

Minor issues

It is not clear what a “Grade A” hospital is (line 58)

The analysis could be simplified.  With sensitivity, specificity, and prevalence, there is no need for the other calculations – these could all be determined by an interested reader. However, I do not know what the consistency rate is.

It is unclear what table 1 adds.  More information about the different regions and the challenges of screening might be helpful.  Table 1 and its associated figure could be reduced and instead consider focus on whether there were important differences across the sites. 

Much of the discussion repeats the results.  A greater focus on interpreting the findings would be helpful.

Author Response

Reviewer #2:

  1. This report describes newborn screening for congenital heart disease (CHD) from 2019-2023 using pulse-oximetry and the physical exam for murmurs. This is referred to as the “dual-index method.” Much of this report lumps together all types of CHD. However, increasing the focus on CCHD would make it line up better with other studies in the field and focus on those who really benefit from early detection. The distinction is addressed later in the report, but this should be woven in throughout.

Response: Thanks for your comments and suggestions. We agree that distinguishing between different types of congenital heart disease (CHD), particularly critical CHD (CCHD) and major CHD, is important for aligning our findings with other studies in the field and emphasizing the subgroup of infants who would benefit most from early detection.

       However, one of the objectives of our study was to report the progress and result of the newborn screening program for all types of CHD in the south Shanghai. This program is designed not only to identify CCHD but also to detect all types of CHD in newborns, ensuring that early diagnosis and appropriate treatment are provided for every type of CHD. Therefore, while we have placed emphasis on life-threatening major CHD, the scope of our study also includes other types of CHD to reflect the comprehensive nature of the screening program.

       We recognize the importance of highlighting CCHD, particularly because of its life-threatening nature and the clear benefits of early intervention. In response to your comment, we have revised the manuscript to weave the distinction between CCHD and other types of CHD throughout the manuscript, providing a more nuanced interpretation of our findings.

  1. Historically, pulse-oximetry was added to the physical exam, which had been the standard of care, to identify critical CHD (CCHD). Using the term “dual-index” seems to complicate matters. However, this could be justified if there was something special about the exam for murmurs (e.g., a standardized approach with little inter- and intra-observer variability).

Response: Thanks for your comments and suggestions. We understand your concern that the terminology “dual-index” might seem to complicate matters, especially given that pulse oximetry and cardiac murmur auscultation have traditionally been used in the identification of CCHD. The use of the terminology "dual-index" in our manuscript is to align with both our previous studies and the Newborn Screening Program for CHD in Shanghai, thus providing continuity in the understanding of this screening method. In response to your point, we would like to emphasize that all screening personnel involved in this study underwent standardized training, which was designed to ensure that the cardiac murmur auscultation was performed consistently with minimal inter- and intra-observer variability. This training process was crucial to maintaining the reliability and validity of the murmur auscultation as part of the "dual-index" method. We have now included more detailed information about this standardized training protocol in the method section to further clarify this point (Page 3, Line 100-102).

  1. It is unclear what this study adds to what has already been described about newborn screening for CCHD. One missed opportunity is to describe implementation issues related to the COVID-19 pandemic. Given the restrictions, I am sure that this important work was challenging. There could be important lessons. The discussion makes describing the progress; however, the analysis is more about the overall findings.

Response: Thanks for your suggestions and raising an important point regarding the contribution of our study in the context of existing literature on newborn screening for CCHD. First, we would like to emphasize that our study adds to the existing body of research by providing up-to-date data from Shanghai on the effectiveness of newborn screening for CCHD the dual-index method. This study underscores the importance of using the dual-index method rather than relying on POX alone and highlights critical symptoms to be aware of for early detection. Additionally, a major focus of our research is evaluating the continued impact of the dual-index method on CHD detection in clinical practice, including an assessment of postnatally diagnosed CHD cases undetected by the dual-index method. Importantly, our current analysis provides a more in-depth evaluation of false-negative cases identified through the dual-index method, with an emphasis on which cases were missed and how these cases were detected post-discharge (Page 2, Line 66-71).

Regarding your point about the COVID-19 pandemic and the potential challenges it posed, we agree that this is an important consideration. Throughout the pandemic, Shanghai maintained essential healthcare services, including neonatal screening, following strict safety protocols to protect both healthcare providers and families. While the the COVID-19 pandemic brought about several challenges to the healthcare system worldwide, including altered healthcare delivery models and the need for enhanced infection control measures, this current study and our previous study demonstrated that the COVID-19 restrictions did not affect the ability to deliver neonatal screenings for CCHD, and the screening for congenital heart disease continued as usual. We have added it in our manuscript (Page 3, Line 115-118; Page 8, Line 283-286).

Minor issues

  1. It is not clear what a “Grade A” hospital is (line 58)

Response: Thanks for your comments. In China, a "Grade A" hospital refers to a ​​Class A Tertiary Hospital​​, which represents the highest tier within the country's hospital classification system. Chinese hospitals are categorized into three levels (primary, secondary, tertiary) based on their ​​service capacity, infrastructure, and clinical expertise. Tertiary hospitals are regional medical centers equipped to handle complex cases, conduct advanced research, and provide specialized training. Within this tier, hospitals are further graded as ​​A, B, or C​​, with "Grade A" signifying the highest quality standards. We have already clarified the definition of a "Grade A" hospital in our manuscript to ensure that readers fully understand it (Page 2, Line 59-60).

  1. The analysis could be simplified. With sensitivity, specificity, and prevalence, there is no need for the other calculations – these could all be determined by an interested reader. However, I do not know what the consistency rate is.

Response: Thanks for your comments. The reason we included additional diagnostic indicators beyond sensitivity, specificity, and prevalence, such as the positive predictive value, negative predictive value, consistency rate, was to align our study with previously published literature. This facilitates comparison across different studies and helps place our findings within the broader research context.

Regarding the "consistency rate," this is commonly used in screening studies. It refers to the proportion of cases in which the screening test result agrees with the gold standard diagnosis, i.e., (true positives + true negatives) divided by the total number of subjects. It serves as a general measure of overall agreement. If necessary, we are willing to revise the manuscript to focus on the core indicators (such as sensitivity, specificity, and prevalence) while potentially moving supplemental metrics to an appendix or supplementary file, or removing them entirely to enhance clarity.

  1. It is unclear what table 1 adds. More information about the different regions and the challenges of screening might be helpful. Table 1 and its associated figure could be reduced and instead consider focus on whether there were important differences across the sites.

Response: Thanks for your comments and suggestions. We have revised the Table 1 accordingly to enhance its value and clarity (Page 5).

Much of the discussion repeats the results. A greater focus on interpreting the findings would be helpful.

Response: Thanks for your comments and suggestions. We have substantially revised the Discussion to place greater emphasis on the interpretation and implications of our findings, rather than simply restating them.

Round 2

Reviewer 1 Report

Comments and Suggestions for Authors

The authors have addressed the concerns I raised during the initial review.   These changes strengthen the manuscript.  The authors have added additional information regarding COVID 19 that are also beneficial.